# Omitted Variable Bias in Language Models Under Distribution Shift

Victoria Lin [1]   Louis-Philippe Morency [1]   Eli Ben-Michael [1]

## Abstract

Despite their impressive performance on a wide variety of tasks, modern language models remain susceptible to distribution shifts, exhibiting brittle behavior when evaluated on data that differs in distribution from their training data. In this paper, we describe how distribution shifts in language models can be separated into *observable* and *unobservable* components, and we discuss how established approaches for dealing with distribution shift address only the former. Importantly, we identify that the resulting *omitted variable bias* from unobserved variables can compromise both evaluation and optimization in language models. To address this challenge, we introduce a framework that maps the strength of the omitted variables to bounds on the *worst-case generalization performance* of language models under distribution shift. In empirical experiments, we show that using these bounds directly in language model evaluation and optimization provides more principled measures of out-of-distribution performance, improves true out-of-distribution performance relative to standard distribution shift adjustment methods, and further enables inference about the strength of the omitted variables when target distribution labels are available.

## 1. Introduction

For language models to be widely useful in real-world applications, they must remain robust, reliable, and performant across data distributions and contexts. However, even for modern large language models (LLMs) capable of impressive performance on diverse tasks, the challenge of *distribution shift* remains. When evaluated on data that differs in distribution from their training data, LLMs often exhibit brittle behavior, such as difficulty answering questions con-

taining simple mutations that do not appear in training (Xu et al., 2025; Huang et al., 2025) or failures in reasoning given counterfactual information that alters the question's world state (González & Nori, 2024; Hüyük et al., 2025).

Established approaches for dealing with distribution shift in language models account for shifts that are *observable* to language models—that is, shifts resulting from variables that models are able to capture from the text. We place a novel focus on a component of distribution shift that these existing methods fail to address: the *unobservable* component of distribution shift, or shifts over variables that cannot be captured directly by language models. This non-observability may occur for two reasons. (i) The shift may occur over variables that are external to the text, such as attributes of the individuals who wrote or labeled the texts. These variables typically remain unmeasured in observational text settings. (ii) The shift may occur over information that is contained in the *text* but not in a model's *representation* of the text. While this information is observable to a human reader, it is not observable to a language model. This type of information loss is significant, since it almost inevitably occurs in language data as nearly infinite-dimensional, unstructured raw text is reduced to a lower-dimensional numerical form that can be used by models.

In this paper, we first identify that the *omitted variable bias* (OVB) arising from these unobserved variables can compromise the evaluation and optimization of language models under distribution shift. In evaluation, failure to account for omitted variables can lead to overly optimistic performance estimates in the target domain; in optimization, it can yield models that remain brittle despite adjustment for observable sources of distribution shift.

To address these challenges, we introduce a framework that maps the strength of the omitted variables to a bound on a language model's generalization performance. Under this framework, although the omitted variables themselves are unobserved, their potential influence can be benchmarked and used to identify a set of plausible target distributions over which the *worst-case generalization performance* of a language model can be defined.

This worst-case generalization bound gives rise to three primary contributions, which we demonstrate empirically. First, the bound provides a more principled and robust met-

---

[1]Carnegie Mellon University, Pittsburgh, PA, USA. Correspondence to: Victoria Lin <vlin2@andrew.cmu.edu>.

*Proceedings of the 43rd International Conference on Machine Learning*, Seoul, South Korea. PMLR 306, 2026. Copyright 2026 by the author(s).

ric for evaluating generalization performance in the absence of target distribution labels, improving on standard adjustment objectives that account only for observed distribution shift. Second, the bound can be directly optimized to produce models that are explicitly more robust to unobserved sources of shift and that generalize more reliably to the target distribution. Third, when target labels *are* available and true test performance can be computed, the bound can be used to infer the strength of the omitted variables for a given model under distribution shift, offering interpretability to models that are otherwise opaque.

## 2. Related Work

**Domain adaptation and distribution shift.** Model performance degradation resulting from differences between source (training) and target (test) distributions has long been studied in the machine learning literature. These distribution shifts arise from a variety of factors, such as differing domains or disparities among subgroups (Arjovsky et al., 2020; Yang et al., 2023). Even for modern LLMs, selection biases when collecting data for preference alignment may skew the distribution of human feedback during post-training (Lin et al., 2024); and recent work on reasoning suggests that LLMs' strong performance may be at least partially attributed to memorization of their training data, as even minor perturbations in test distributions can elicit large drops in performance (Xu et al., 2025; Huang et al., 2025).

One of the most common and portable methods for addressing distribution shift is *importance weighting*, which reweights training samples to match the target distribution via a density ratio. Importance-weighted estimators are often combined with outcome models to form *doubly robust* estimators that are more robust to misspecification and estimation error (Robins et al., 1994). These methods are widely used to correct for covariate and label shift across prediction, recommendation, and reinforcement learning tasks (Byrd & Lipton, 2019; Kallus et al., 2022; Kim et al., 2022; Lin et al., 2024). However, when relevant variables are omitted, bias can arise in both the density ratio and outcome model, threatening the validity of these adjustments.

**Omitted variable bias.** In practice, OVB can be difficult to estimate, as computing it directly requires knowing which variables have been omitted from the unknown full set of variables. Moreover, a particular challenge for language models is that the "full set of variables" corresponds to the raw text used as input—but any method modeling text still requires that it be represented numerically, and so the estimator loses access to the omitted variables.

Due to the difficulties of computing OVB directly, a long history of work has set out to provide bounds on the degree of OVB under various (generally linear) modeling assump-

tions (Goldberger, 1991; Frank, 2000; Angrist & Pischke, 2009; Oster, 2019; Cinelli & Hazlett, 2019). To address the limitations of these strong assumptions, recent work from Chernozhukov et al. (2024) establishes bounds on the OVB of causal parameters even when using nonparametric models. Subsequent work from Lin et al. (2025) extends these bounds to a language setting where the goal is to estimate causal effects from language. However, because most language tasks are not explicitly causal, the question of how to extend bounds and other theoretical properties of omitted variables to language models becomes less clear. Our work addresses this challenge by mapping the strength of omitted variables to bounds on the generalization performance of language models under a general distribution shift setting.

## 3. Proposed Approach

### 3.1. A General Form for Doubly Robust Losses

We consider a distribution shift setting where the source distribution $P$, from which training data is derived, may differ from the target distribution $Q$. Then the *generalization performance* of a model $f$ under this setting is given by its performance in the target distribution, or

$$\mathcal{L} = \mathbb{E}_Q[\ell(Y, X; f)],$$

where $X$ are texts, $Y$ are labels, and $\ell$ is the model loss.

To learn a model with good generalization performance, the goal is to minimize this quantity. In a distribution shift setting, however, labels are available only for the source distribution $P$ and not the target distribution $Q$. That is, we observe a labeled dataset from $P$:

$$D_P = \{(X_i, Y_i) \sim P\},\ i \in [n] \quad X_i \in \mathcal{X}, Y_i \in \mathbb{R},$$

and an unlabeled dataset from $Q$.

$$D_Q = \{X_j \sim Q\},\ j \in [m] \quad X_j \in \mathcal{X}.$$

**Identification.** Since $\mathcal{L}$ is written in terms of the unobserved labels $Y \sim Q$, it cannot be directly computed. We will consider a setting where it is possible in principle to identify $\mathcal{L}$ in terms of the observed distribution $P$ given access to the text $X$ and potentially additional covariates $Z$ that are external to the text and that we do not observe.

The following assumptions are standard assumptions for domain shift, if we had access to these additional external values $Z$. (i) **Covariate shift.** We assume a covariate shift setting where—given access to the full set of variables—the relation $\mathbb{E}_P[\ell(Y, X; f) \mid X, Z] = \mathbb{E}_Q[\ell(Y, X; f) \mid X, Z]$ holds. (ii) **Overlap.** We assume overlap between the source and target distributions: $\frac{dQ}{dP}(X, Z) = \frac{\Pr_Q(X,Z)}{\Pr_P(X,Z)}$ is finite for

all pairs of $X$ and $Z$. In intuitive terms, this assumption ensures that any text with a non-zero probability of appearing in $Q$ has also a non-zero probability of appearing in $P$.

If we had access to the unobserved external variables $Z$, then letting $\alpha(X, Z) = \frac{\Pr_Q(X, Z)}{\Pr_P(X, Z)}$ be the Riesz representer, a common approach for distribution shift adjustment is to compute the *importance-weighted* objective:

$$\mathcal{L}_{\text{IPW}} = \mathbb{E}_P \left[ \alpha(X, Z) \ell(Y, X; f) \right].$$

The addition of an outcome model $g(h(X), X, Z; f) = \mathbb{E}[\ell(Y, X; f) \mid X, Z]$ then yields the following *doubly robust* objective:

$$\mathcal{L}_{\text{DR}} = \mathbb{E}_P \left[ \alpha(X, Z)(\ell(Y, X; f) - g(X, Z; f)) \right] \\ + \mathbb{E}_Q[g(X, Z; f)].$$

Note that here $g$ is a predictor of the loss $\ell$ itself rather than the label $Y$. Both $\mathcal{L}_{\text{IPW}}$ and $\mathcal{L}_{\text{DR}}$ are mathematically equivalent to the original $\mathcal{L}$ but allow the loss to be computed over the observed data $D_P$. $\mathcal{L}_{\text{DR}}$ further confers robustness against misspecification of the density ratio $\frac{\Pr_Q}{\Pr_P}$ as long as the outcome model $g$ is correct, or vice versa. Derivations for these objectives are found in Appendix A.1.

**Estimation.** In practice, any estimator of the doubly robust objective will have access only to variables observable to language models. Notably, in the language model setting, these observable variables correspond only to $h(X)$, the language representation of the text $X$, where $h(\cdot) : \mathcal{X} \to \mathbb{R}^d$, and do not include either the full text or the unobserved external variables $Z$.

Following the terminology used in Chernozhukov et al. (2024) to refer to models learned over partial feature sets, we first define the "short" outcome model $g(h(X); f) = \mathbb{E}_P[\ell(Y, X; f) \mid h(X)]$ and "short" Riesz representer $\alpha(h(X)) = \frac{\Pr_Q(h(X))}{\Pr_P(h(X))}$, which are learned given only the text representation $h(X)$. These are contrasted with the "long" outcome model and "long" Riesz representer, which are learned over the full text $X$ and unknown external variables $Z$ (and therefore also implicitly over the representation $h(X)$, which is a deterministic function of $X$).

Together, these yield an estimator of the short doubly robust objective $\mathcal{L}_{\text{DR}_s}$ that replaces the long models with the short models:

$$\widehat{\mathcal{L}}_{\text{DR}_s} = \widehat{\mathbb{E}}_P \left[ \widehat{\alpha}(h(X))(\ell(Y, X; f) - \widehat{g}(h(X); f) \right] \\ + \widehat{\mathbb{E}}_Q[\widehat{g}(h(X); f)],$$

where $\widehat{g}$ is estimated on a held-out sample from $P$, and $\widehat{\alpha}$ is estimated on a held-out sample of texts from both $P$ and $Q$. $\widehat{\mathbb{E}}_P$ represents the sample average in the dataset drawn from the source distribution $P$.

Adjustment using only the model-observable variables $h(X)$ accounts for only the first component of distribution shift. In essence, it assumes that no information in the text $X$ other than the representation $h(X)$ is systematically different between the source and the target. It also assumes that the unobserved external variables $Z$ are not systematically different. Concretely, this assumes that the covariate shift assumption holds when conditioning only on $h(X)$, rather than the full set of variables $X$ and $Z$.

When this assumption fails and there are systemic differences between the source and target distributions not captured solely by the representation $h(X)$, OVB is introduced in the short doubly robust objective $\mathcal{L}_{\text{DR}_s}$, and the estimator $\widehat{\mathcal{L}}_{\text{DR}_s}$ may no longer accurately reflect generalization performance in the target distribution $Q$. As a result, models learned using this objective may insufficiently account for distribution shift and perform poorly when deployed outside their training domain.

### 3.2. Worst-Case Generalization Performance

Without access to the full covariate set $(X, Z)$, unbiased estimation of $\mathcal{L}_{\text{DR}}$ is no longer possible under standard identification assumptions. Relying only on the representation $h(X)$ yields a partially identified generalization objective whose value depends on unobserved variables omitted from the distribution shift adjustment set. We propose to account for this uncertainty by incorporating the omitted variable bias induced by the unobserved covariates (Chernozhukov et al., 2024).

Without full knowledge of the covariates, the observed data define a set of plausible target distributions over which the model's performance may vary. To account for this uncertainty, we adopt a distributionally robust optimization (DRO) framework (Duchi & Namkoong, 2021) to define the doubly robust objective under the worst-case distribution with respect to OVB within this set of plausible targets. This approach yields a *bound on the model's generalization performance* that provides robustness guarantees under unobserved distribution shift. We therefore expect that models evaluated using this worst-case objective will more accurately reflect performance in the target distribution, and models learned using this objective will generalize more reliably to the target distribution than those trained using the standard doubly robust objective.

**OVB definition.** Extending Chernozhukov et al. (2024), the OVB of $\mathcal{L}_{\text{DR}_s}$ is bounded as:

$$\underbrace{|\mathcal{L}_{\text{DR}_s} - \mathcal{L}_{\text{DR}}|^2}_{\text{OVB}} \leq \rho^2 C_Y^2 C_D^2 \sigma^2 \nu^2,$$

where the *fidelity* $\sigma^2$ and the *overlap* $\nu^2$ are identifiable

directly from the data as

$$\sigma(f)^2 = \mathbb{E}_P[(\ell(Y, X; f) - g(h(X); f))^2],$$
$$\nu^2 = \mathbb{E}_P[\alpha(h(X))^2].$$

The fidelity measure $\sigma^2$ indicates how well the short outcome model predicts the loss, while the overlap measure $\nu^2$ indicates how well the overlap assumption between $P$ and $Q$ is fulfilled by the short Riesz representer.

The OVB bound furthers depend on the parameters $C_Y$, $C_D$, and $\rho$, where $C_Y$ and $C_D$ denote the explanatory power of the omitted variables toward the outcome model and Riesz representer, respectively, and $\rho$ is the degree of confounding present. These are defined as

$$C_Y(f)^2 = \frac{\mathbb{E}_P[(g(X, Z; f) - g(h(X); f))^2]}{\mathbb{E}_P[(\ell(Y, X; f) - g(h(X); f))^2]},$$
$$C_D^2 = \frac{\mathbb{E}_P[\alpha(X, Z)^2] - \mathbb{E}_P[\alpha(h(X))^2]}{\mathbb{E}_P[\alpha(h(X))^2]},$$
$$\rho(f) = \text{Corr}^2(g(X, Z; f) - g(h(X); f),$$
$$\alpha(X, Z) - \alpha(h(X))).$$

Because the long outcome model $g(X, Z; f)$ and the long Riesz representer $\alpha(X, Z)$ are not known (except in an oracle setting), these quantities are not identified directly from the data. Instead, they serve as sensitivity parameters to characterize the level of worst-case OVB.

**Worst-case objective.** We establish that these OVB bounds can now be used to construct an uncertainty set for a DRO problem. Under these bounds, the observed data admit a family of target distributions characterized by $C_Y$, $C_D$, and $\rho$. Letting $f \in \mathcal{F}$ denote a model from a fixed hypothesis class, we define:

$$C_Y^{\max} := \sup_{f \in \mathcal{F}} C_Y(f), \ \rho^{\max} := \sup_{f \in \mathcal{F}} \rho(f), \ C_D^{\max} := C_D.$$

Then we define the set of possible target distributions as

$$\mathcal{Q} = \left\{ \tilde{Q} : C_Y \leq C_Y^{\max}, C_D \leq C_D^{\max}, \rho \leq \rho^{\max} \right\}.$$

As we mention previously, $C_Y$, $C_D$, and $\rho$ (and consequently $C_Y^{\max}$, $C_D^{\max}$, and $\rho^{\max}$) cannot be identified directly from the data—but they *can* be benchmarked to observed data. The resulting benchmarked values can then be used to define a plausible range for these sensitivity parameters, as we later show in our empirical experiments.

Then for a fixed model $f \in \mathcal{F}$, the worst-case generalization performance[1] over this set is given by

$$\sup_{\tilde{Q} \in \mathcal{Q}} \mathbb{E}_{\tilde{Q}}[\ell(Y, X; f)] = \mathcal{L}_{\text{DR}} + \rho^{\max} C_Y^{\max} C_D^{\max} \sigma \nu.$$

Evaluating models with this performance metric directly leads to our first contribution: the ability to robustly evaluate the generalization performance of language models in the target domain.

Next, by minimizing this quantity, i.e., solving

$$\min_{f \in \mathcal{F}} \sup_{\tilde{Q} \in \mathcal{Q}} \mathbb{E}_{\tilde{Q}}[\ell(Y, X; f)],$$

we arrive at our second contribution: a model optimized in this way should generalize more reliably to the target distribution because it explicitly accounts for the uncertainty introduced by unobserved sources of distribution shift.

Finally, the sensitivity parameters themselves bring us to our third contribution. If a model's performance on the target distribution can be computed directly—for example, if we have access to some target labels—then it is possible to infer an empirical upper bound on $\rho$, $C_Y$, and $C_D$ by tuning the sensitivity parameters until the worst-case generalization performance matches the observed test performance. Comparing these sensitivity parameters among language models provides useful insight into the strength of the omitted variables for each model: larger values of the sensitivity parameters correspond to more explanatory power coming from the omitted variables. This in turn signals whether or not a model has captured the concepts that distinguish the source and target distribution, conferring a degree of interpretability on models that otherwise might have none.

## 4. GLM Log-Likelihood Losses

Our formulation can be used to bound worst-case generalization performance for any generic task loss. However, it requires that the model $g$ predict the loss itself as the conditional outcome rather than the label. Because the loss depends on the language model $f$, this means the nuisance model must simulate or predict the predictions of the primary model, meaning that a new $g$ must be learned for every update step. For modern LLM tasks like text generation, this is challenging, computationally demanding, and can introduce optimization instability during training.

In this section, we propose a highly tractable restructuring of this objective for the family of generalized linear model (GLM) log-likelihood losses. This class of losses includes tasks considered to be core to natural language processing,

---

[1]Confidence intervals on the OVB can also be computed following Chernozhukov et al. (2024). These can be used directly to define a confidence bound on worst-case generalization performance.

*Table 1.* NLP tasks with GLM log-likelihood losses and their correspondence to language model outputs $f(x)$.

| | $\eta$ | $b(\eta)$ | Label | Mean parameter |
|---|---|---|---|---|
| **Regression** | $f(x) \in \mathbb{R}$ | $\frac{1}{2}\eta^2$ | $Y \in \mathbb{R}$ | $\mu = \eta$ |
| **Binary classification** | $f(x) \in \mathbb{R}$ (logit) | $\log(1 + \exp(\eta))$ | $Y \in \{0,1\}$ | $p = \sigma(\eta)$ |
| **Multiclass classification** | $f(x) \in \mathbb{R}^K$ (logits) | $\log \sum_{j=1}^K \exp(\eta_j)$ | $Y \in \{0,1\}^K$ | $p_j = \frac{\exp(\eta_j)}{\sum_{k \in K} \exp(\eta_k)}$ |
| **Text generation** | $f^{(t)}(x) \in \mathbb{R}^K$ (logits at step $t$) | $\sum_{t=1}^T \log \sum_{j=1}^K \exp(\eta_j^{(t)})$ | $Y^{(t)} \in \{0,1\}^K$ | $p_j^{(t)} = \frac{\exp(\eta_j^{(t)})}{\sum_{k \in K} \exp(\eta_k^{(t)})}$ |

such as regression, classification, and text generation. Due to their log-linear structure, these losses allow for an alternative implementation of the bound on generalization performance where the outcome model simply predicts the label. This results in a much easier optimization problem that is widely useful for many common tasks.

Derivations and technical results for this section are deferred to Appendix A.2 and B.

### 4.1. A Worst-Case Objective for GLM Losses

Following the generic form of a GLM negative log-likelihood, let

$$\ell(Y, X; \eta, f) = -(Y \cdot \eta(X; f) - b(\eta(X; f)) + c(Y)),$$

where $\eta(\cdot)$ is the natural parameter, $b(\cdot)$ is the log-partition function, and $c(Y)$ is a constant with respect to optimization and therefore can be ignored. Then we have the losses:

$$\mathcal{L} = \mathbb{E}_Q[b(\eta(X; f)) - Y \cdot \eta(X; f)],$$
$$\mathcal{L}_{\text{IPW}} = \mathbb{E}_Q[b(\eta(X; f))] - \mathbb{E}_P\left[\frac{\text{Pr}_Q(X, Z)}{\text{Pr}_P(X, Z)}\eta(X; f)Y\right].$$

This yields an alternative decomposition of the doubly robust objective in which the portion of the loss that is dependent on $f$ is contained in the Riesz representer $\alpha$ rather than the outcome model $g$:

$$\alpha(X, Z; f) = \frac{\text{Pr}_Q(X, Z)}{\text{Pr}_P(X, Z)}\eta(X; f).$$

The GLM log-likelihood loss allows us to define the conditional outcome as simply the label $Y$, and we can relax our covariate shift assumption compared to the requirement for the general form in Section 3.1. Here, we assume simply that $\mathbb{E}_P[Y \mid X, Z] = \mathbb{E}_Q[Y \mid X, Z]$. Letting $g$ now be a predictor of the label $Y$ rather than the loss, i.e., $g(X, Z) = \mathbb{E}[Y \mid X, Z]$, we have

$$\mathcal{L}_{\text{DR}} = \mathbb{E}_Q[b(\eta(X; f)) - \eta(X; f)g(X, Z)]$$
$$- \mathbb{E}_P[\alpha(X, Z; f)(Y - g(X, Z))],$$

and as before, the worst-case generalization performance of model $f$ over the plausible target distribution set $\mathcal{Q}$ is

$$\sup_{\tilde{Q} \in \mathcal{Q}} \mathbb{E}_{\tilde{Q}}[\ell(Y, X; f)] = \mathcal{L}_{\text{DR}} + \rho^{\max} C_Y^{\max} C_D^{\max} \sigma\nu.$$

**Task-specific parameters.** From this general form, the loss functions and worst-case task objectives for regression, classification, and text generation are easily obtained by using the correct natural parameter $\eta$ and log-partition function $b$ for each task (we identify these in Table 1), then substituting into the GLM worst-case objective.

### 4.2. From Long to Short

Under the alternative decomposition of the doubly robust objective, the short outcome model and Riesz representer are now given by $g(h(X)) = \mathbb{E}_P[Y \mid h(X)]$ and $\alpha(h(X); f) = \frac{\text{Pr}_Q(h(X))}{\text{Pr}_P(h(X))}\eta(h(X); f)$, respectively. From these, the fidelity $\sigma^2$ and overlap $\nu^2$ follow:

$$\sigma^2 = \mathbb{E}_P[(Y - g(h(X)))^2], \quad \nu^2 = \mathbb{E}_P[\alpha(h(X); f)],$$

and all other parameters associated with the worst-case objective are defined as before.

## 5. Experiments

In our experiments, we study three use cases of the worst-case generalization bound aligned with our contributions. First, we use the bound for evaluation and show that ignoring unobserved variables leads to overly optimistic estimates of target performance. Second, we demonstrate how the bound can quantify the strength of omitted variables even for a black-box commercial LLM on a complex reasoning task. Finally, across multiple datasets and tasks, we show that optimizing models for worst-case generalization improves out-of-distribution performance relative to standard doubly robust methods that assume no missing information.

### 5.1. Datasets

**Math Reasoning (text generation, real-world setting).** Our Math Reasoning dataset pairs the original MATH

dataset (Hendrycks et al., 2021) and a recent variation, MATH-Perturb (Huang et al., 2025), both consisting of problems from high school math competitions. MATH-Perturb perturbs existing problems in the MATH dataset, with two versions—MATH-P-Hard and MATH-P-Simple—that do and do not change the underlying logic of the original question, respectively. We treat MATH as the source distribution and each variant of MATH-Perturb as a target distribution. This dataset is particularly appropriate for evaluating OVB in LLMs, since MATH-Perturb was developed in direct response to concerns that LLMs' good benchmark performance on MATH is due to memorization of the dataset in training, and its authors show that LLMs perform significantly worse on MATH-Perturb than on MATH.

**Amazon (regression, semi-synthetic setting).** The Amazon dataset (McAuley & Leskovec, 2013) consists of customer product reviews, each associated with "helpfulness" votes from other customers. We take reviews of musical instruments and office products as our respective source and target distributions. To ensure control of the omitted variables, we encode the reviews with the LIWC lexicon (Pennebaker et al., 2015) and generate a new continuous semi-synthetic label $Y$ using the 10 LIWC features most predictive of the helpful vote count. From this, variables can be intentionally omitted by restricting a model's access to only a subset of these 10 features at training time.

**EmoBank (regression, real-world setting).** The EmoBank dataset (Buechel & Hahn, 2017) consists of sentences labeled by human annotators according to their valence, or the positivity or negativity of the text, on a 5-point scale ($Y$). We create two splits of this dataset over their *writer-intended* valence, taking texts with high and low writer-intended valence as the source and target distributions, respectively. Importantly, writer-intended valence differs from the label, which is the annotator's *perceived* valence, but is correlated with it. This creates a natural data setting where an obvious variable likely to induce differences between the source and target distributions—the writer's intent—is not fully observable.

**Hate Speech (classification, real-world setting).** The Hate Speech dataset (Qian et al., 2019) consists of user comments from the social media sites Reddit and Gab. Each comment is labeled by a human annotator with a binary indicator of hate speech ($Y$). We use Reddit as the source distribution and Gab as the target distribution. This dataset again allows us to evaluate a natural setting where OVB may arise, as some variables explaining differences between Reddit and Gab comments may not be observable (e.g., factors that impact both a user's choice of site to use and the type of content the user tends to post).

**Wild-Time (classification, real-world setting).** A highly relevant form of distribution shift—particularly for modern LLMs that are trained and evaluated on constantly evolving data—is temporal drift, or distribution shift over time. The Huffpost dataset from the Wild-Time temporal drift benchmark (Yao et al., 2022) contains headlines of news articles published in the Huffington Post between the years 2012 and 2018, where each article is tagged with a news category $Y$. In our experiments, we consider headlines in the Black Voices and Business categories (by default the first two categories in the dataset). We use headlines from the year 2012 as the source distribution and iterate over data splits from the years 2013-2018 as the target distributions.

## 5.2. Language Model Evaluation and Measuring Omitted Variables

In our first two sets of experiments, we use the worst-case generalization bound to evaluate model performance and measure the strength of omitted variables on the complex Math Reasoning dataset. We consider a setting where we have access to labels in the target distribution.

We take pre-trained GPT-4.1 (OpenAI et al., 2024) as our language model $f$, with an expectation based on Huang et al. (2025) that it has already seen the source distribution MATH in its training data but not the target distribution MATH-Perturb. To fit the nuisance parameters $\widehat{g}$ and $\widehat{\alpha}$, we use simple models based on the LIWC and Empath (Fast et al., 2016) lexicons; earlier-generation transformers BERT (Devlin et al., 2019), RoBERTa (Liu et al., 2019), and MiniLM (Wang et al., 2020); and GPT-4.1 itself.

For each model-specific generalization bound, we estimate the corresponding upper bound on the OVB sensitivity parameters, $\rho C_Y C_D$, by tuning them until the bound intersects the true test performance. We also compute $\rho C_Y C_D$ at the intersection between the confidence interval of worst-case generalization performance and the true test performance. Together, these intersections define a reasonable range for the sensitivity parameters, which characterize the strength of the omitted variables. Comparing these values across models therefore indicates the degree to which each model is susceptible to OVB from unobservable distribution shifts.

## 5.3. Language Model Optimization

**Semi-synthetic setting.** We recall that in the semi-synthetic Amazon setting, we generate the label $Y$ from 10 known LIWC features and omit variables by providing incomplete subsets of those 10 features to models at train time. Because both the "long" and "short" feature sets are known, all values required to compute the worst-case generalization bound—including the true sensitivity parameters—can be identified directly from the data. For 100 iterations, we omit a random number of random features and train a model $f$

using the worst-case generalization bound, then compute the test performance of the model on the target distribution.

**Real-world settings.** In our real-world dataset evaluations, we use a variety of language representations that reflect the complexity of each setting and task. For the EmoBank dataset, texts are encoded using LIWC, and for the Hate Speech dataset, comments are encoded using the SenteCon lexicon, which provides more contextual information in its representations compared to LIWC (Lin & Morency, 2023). Finally, for the Wild-Time dataset, headlines are represented as MPNet embeddings, which contain denser information than the lexicon-based representations used for the other two datasets.

In these settings, the true sensitivity parameters are not known and must therefore be chosen by the user. We optimize models for the worst-case generalization bound while iterating over a set of plausible parameter values, where larger values correspond to a more pessimistic bound. Building on precedents from the sensitivity analysis literature (Cinelli & Hazlett, 2019; Oster, 2019), we benchmark these values in the EmoBank and Hate Speech datasets by computing estimates for the sensitivity parameters using a proxy "long" model trained on comments encoded using higher-dimensional representations from the transformer MPNet (Song et al., 2020). When such higher-dimensional representations are unavailable (e.g., for LLMs), analogous benchmarks can be obtained by treating the existing representation as the long model and intentionally omitting features to construct a proxy "short" model, from which sensitivity parameters can be estimated. We use this latter approach for the Wild-Time dataset. In additional experiments on the Amazon dataset, we demonstrate that benchmarking sensitivity parameters with proxy models is effective for identifying plausible values (Appendix D).

**Implementation.** We compare our approach to two baselines: a model optimized using the doubly robust objective $\widehat{\mathcal{L}}_{DR}$ (**DR**), and a model optimized without any distribution shift adjustment (**unadjusted**). We train our nuisance parameters $\widehat{g}$ and $\widehat{\alpha}$ on held-out samples consisting of 30% of the training data. Due to the low-dimensional feature set, we use small MLPs with a single hidden layer of size 100 for both nuisance models, which we implement in `scikit-learn` using default hyperparameters. We use a linear model for the main model $f$ and optimize with `scipy`.

# 6. Results and Discussion

## 6.1. Language Model Evaluation

In Figures 1a and 1b, we plot both the true performance in the target domain and the estimated performance using a standard doubly robust objective, which adjusts only for

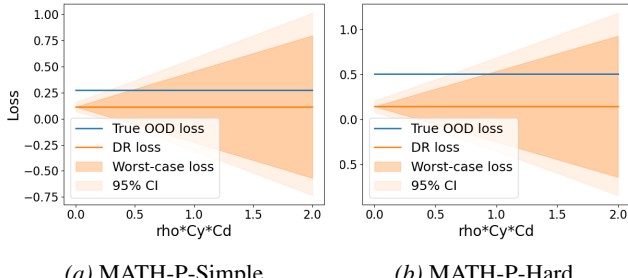

*(a) MATH-P-Simple*      *(b) MATH-P-Hard*

*Figure 1.* Evaluations of true and distribution shift-adjusted performance on MATH-Perturb.

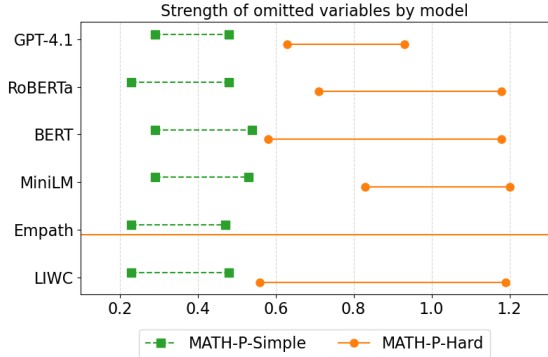

*Figure 2.* Strength of omitted variables under distribution shift in the Math Reasoning dataset.

observed distribution shift. We see that for both MATH-P-Simple and MATH-P-Hard, the doubly robust objective does not recover true performance—though the gap is smaller for the former. This is consistent with expectations, as the problems in MATH-P-Hard contain logic that differs systematically from that of MATH, while MATH-P-Simple varies only on surface-level attributes like the numbers provided in the problem. Importantly, in both cases the doubly robust objective is *optimistic* relative to the true test performance.

Looking to our generalization bound, however, we see that as we increase the assumed strength of the omitted variables, the worst-case performance intersects with the true test performance—indicating that if plausible values are chosen for the sensitivity parameters (e.g., by benchmarking as we previously describe), then accounting for unobserved variables via our bound will provide a more accurate picture of performance in the target distribution.

## 6.2. Strength of Omitted Variables

Building on this setting, we show that our generalization bound can quantify the strength of omitted variables across language models. This evaluation considers two axes: (i) the true degree of difference between source and target distributions, which sets a ceiling on omitted variable strength, and (ii) the extent to which a model captures those differences,

or conversely the strength of the variables it omits.

Across all models, we find that the strength of omitted variables is much larger between MATH and MATH-P-Hard than between MATH and MATH-P-Simple (Figure 2). Again, this is consistent with expectations, since the differences between MATH and MATH-P-Simple are primarily stylistic, while MATH and MATH-P-Hard differ in their fundamental logic. Language models are likely less capable of capturing this higher-order logic.

Moreover, in MATH-P-Simple, all models exhibit similar sensitivity parameter ranges. Because the differences between MATH and MATH-P-Simple are largely surface-level, even simpler language models seem to be able to capture them from the text, and any remaining omitted-variable effects may instead arise from external factors that do not vary across models. In contrast, when MATH-P-Hard is the target distribution, the models exhibit greater differences in their sensitivity parameter ranges. Unsurprisingly, GPT-4.1—the strongest model—seems least affected by omitted variables, with both the tightest range of sensitivity parameters and the lowest upper bound. This appears largely due to GPT-4.1's ability to recognize a significant distributional difference between MATH and MATH-P-Hard, which is demonstrated by its much larger $\widehat{\nu}^2$ (Appendix E, Figure 7). The remaining models have $\widehat{\nu}^2$ close to 1,[2] which indicates that they find almost perfect overlap between MATH and MATH-P-Hard—thereby suggesting that they fail to capture important known differences between the two distributions.

These measures provide a direct way to assess whether a model has learned a specific concept—such as the logic distinguishing MATH from MATH-P-Hard—without targeted probing tasks or relying on performance metrics, which only indirectly reflect concept knowledge and may be confounded by other model capabilities. In this way, OVB-based comparisons offer interpretable insight into black-box models that would otherwise be opaque.

### 6.3. Language Model Optimization

**Amazon.** Over 100 iterations, we compare the test performance of a model optimized with the worst-case generalization bound against the doubly robust and unadjusted baselines (Figure 3). We recall that in this semi-synthetic setting, we have access to the true sensitivity parameters.

We observe that as the product of the true sensitivity parameters increases—i.e., as the true strength of omitted variables increases—our approach increasingly outperforms the

<hr/>

[2]Empath in fact has a large negative $\widehat{\nu}^2$, which is a useful empirical indicator of a potential overlap violation. This violation is reflected in the large amount of OVB. Such empirical detections of limited overlap serve as an essential warning signal of generalization limits.

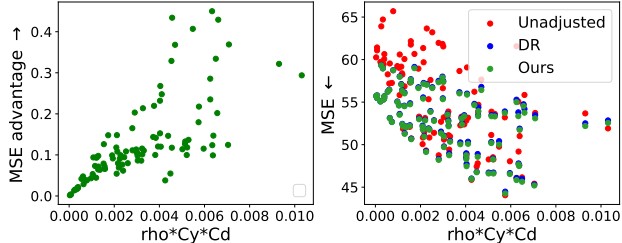

*Figure 3.* Amazon dataset. On the left, we plot the test performance difference (higher is better) between our approach and a model trained with a standard DR objective against the true strength of omitted variables. On the right, we plot the test performance of our approach and both baselines.

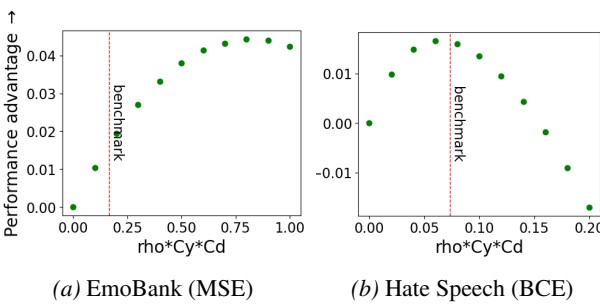

*(a)* EmoBank (MSE)      *(b)* Hate Speech (BCE)

*Figure 4.* Test performance difference (higher is better) between our approach and a model trained with a standard DR objective against assumed strength of omitted variables.

vanilla DR approach with respect to true test performance. This result demonstrates that if sensitivity parameters are appropriately set, (i) our approach is indeed useful for reducing the impact of omitted variables on language model performance, and (ii) the extent to which our approach confers performance advantages over the DR objective increases with the strength of omitted variables.

**EmoBank and Hate Speech.** In real-world settings where the true sensitivity parameters are unknown, we iterate over 11 plausible values of their product—calibrated using the benchmarked value shown in dotted red—to examine how true test performance varies as the optimization objective becomes more conservative (Figures 8a and 8b). When this product is 0, our method reduces to the DR baseline.

Across both datasets, increasing the sensitivity parameters initially improves true test performance relative to vanilla DR, before performance eventually declines. The gain peaks around $\rho C_Y C_D = 0.8$ for EmoBank and around $\rho C_Y C_D = 0.06$ for Hate Speech. These results demonstrate the effectiveness of our approach on real datasets where we have no knowledge of the omitted variables—but also highlight that while accounting for unobserved variables *is* beneficial for model performance under distribution shift, optimizing for an overly pessimistic worst-case gener-

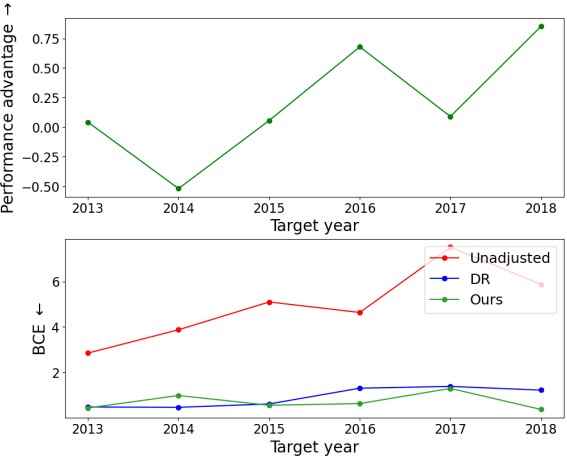

*Figure 5.* Wild-Time dataset over target distribution year. On top, we plot the test performance difference (higher is better) between our approach and a model trained with a standard DR objective. On the bottom, we plot the test performance of our approach and both baselines.

alization bound can be detrimental to model performance.

From these results, we can infer that a relatively large component of distribution shift remains unobserved for EmoBank (that is, the omitted variables are strong), while Hate Speech is much less impacted by unobserved distribution shift. It is possible that most of the distribution shift between Reddit and Gab is already captured in the text representations and therefore observable to language models—or the two distributions may not differ significantly in the first place, which seems plausible since the Reddit portion of the dataset is specifically restricted to subreddits where the user base is expected to be similar to the Gab user base. The distributions of EmoBank, on the other hand, differ over a concrete variable—writer intent—that we know is not directly observable from the data.

**Wild-Time.** In the temporal shift setting, we evaluate distribution shift over multiple years, fixing the 2012 data as the source distribution. For the target distribution corresponding to each year, we benchmark sensitivity parameter values using proxy "short" models learned over representations from the Empath lexicon. We directly set the sensitivity parameters in our worst-case generalization objective to these benchmarked values and evaluate true test performance in the target domain (Figure 5).

First, we note that the overall label proportion generally changes linearly with time, with a decreasing proportion of Business relative to Black Voices (Figure 9). This is consistent with overall societal trends and suggests a general trend of increasing temporal drift over time.

This increasing drift over time is reflected in the optimization results in Figure 5. Here, the error of the unadjusted

baseline shows a general upward trend, suggesting that due to increasing temporal distribution shift, a model trained on 2012 data is increasingly unable to perform on data farther in the future. Notably, the error of the standard DR baseline also shows a (slower) upward trend, indicating that there is also an increase in *unobservable* drift that the standard approach is unable to account for. In contrast, our worst-case objective continues to mitigate these increasing shifts over time, maintaining an advantage over the DR baseline.[3]

## 7. Conclusion

In this paper, we describe a novel unobservable component of distribution shift in language models that can give rise to *omitted variable bias*, compromising model evaluation and optimization. To address this challenge, we present a framework that maps the strength of omitted variables to a bound on the worst-case generalization performance of a language model under distribution shift. We demonstrate three primary contributions that follow from this bound: the ability to (i) more robustly measure models' out-of-distribution performance, (ii) improve true generalization performance relative to standard distribution shift adjustment methods, and (iii) measure the strength of omitted variables when target labels are available.

These contributions point to an interesting line of future work. Although this paper focuses on a distribution shift setting, this approach can be generalized to any language task where the model objective takes or can take a doubly robust form, with the only difference being in how we define the Riesz representer $\alpha$. Therefore, our approach can be used not only to uncover the presence of OVB in other modeling contexts or tasks but can also provide a clear path for mitigating its effects in language models, including LLMs. While directly optimizing an LLM with our proposed objective is computationally intensive, as model weight updates cause the representations $h(X)$ to change at every step, this framework extends naturally to more lightweight approaches like training robust adaptation layers over frozen embeddings. Moreover, common preference optimization methods like PPO-based RLHF and DPO benefit from the more computationally tractable GLM formulation, as the Bradley-Terry preference models on which they rely are GLMs. More broadly, our work suggests a unifying perspective on robustness to omitted variables across a range of language model objectives through their shared doubly robust structure.

---

[3]The performance advantage of our approach is not strictly linear, and for the 2014 data our method anomalously underperforms the DR baseline. This may be due to the sensitivity parameter values used for the worst-case objective. Benchmarked sensitivity parameter values do not always correspond to optimal performance, and there can exist certain distributions where the proxy heuristic yields overly pessimistic parameters that are even detrimental to performance.

## Impact Statement

This work aims to improve the evaluation and optimization of language models under distribution shift by identifying and accounting for the effects of omitted variable bias. By providing tools to account for uncertainty arising from unobserved variables, our framework may help practitioners make more reliable decisions when deploying models in settings where data distributions are imperfectly observed.

Potential positive impacts include more robust model evaluation, reduced overconfidence in reported performance, and improved reliability of language models in real-world applications. At the same time, as with most advances in model evaluation and optimization, our methods can be applied in domains with varying societal consequences depending on the deployment context. We believe the ethical and societal implications of this work are consistent with those commonly associated with advancing robustness and generalization in machine learning, and we do not foresee specific negative impacts unique to this contribution.

## Acknowledgments

This material is based upon work partially supported by the National Institutes of Health (federal award ID numbers R01MH125740, R01MH132225, and U01MH136535), the National Institute of Standards and Technology (federal award ID number 6ONANB24D231), and the Carnegie Mellon University AI Measurement Science and Engineering Center (AIMSEC). Victoria Lin is partially supported by a Meta Research PhD Fellowship. Any opinions, findings, conclusions, or recommendations expressed in this material are those of the author(s) and do not necessarily reflect the views of the sponsors, and no official endorsement should be inferred.

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

# A. Derivations

### A.1. General Doubly Robust Objective

Recall the covariate shift assumption $\mathbb{E}_Q[\ell(Y, X; f) \mid X, Z] = \mathbb{E}_Q[\ell(Y, X; f) \mid X, Z]$ and that $g(X, Z; f) = \mathbb{E}[\ell(Y, X; f) \mid X, Z]$:

$$
\begin{aligned}
\mathcal{L} &= \mathbb{E}_Q[\ell(Y, X; f)] \\
&= \mathbb{E}_Q\left[\mathbb{E}_Q[\ell(Y, X; f) \mid X, Z]\right] \\
&= \mathbb{E}_Q\left[\mathbb{E}_P[\ell(Y, X; f) \mid X, Z]\right] \\
&= \mathbb{E}_Q\left[g(X, Z; f)\right].
\end{aligned}
$$

And

$$
\begin{aligned}
\mathcal{L} &= \mathbb{E}_Q[\ell(Y, X; f)] \\
&= \mathbb{E}_P\left[\frac{dQ}{dP}(X, Z)\ell(Y, X; f)\right] \\
&= \mathcal{L}_{\text{IPW}}.
\end{aligned}
$$

So

$$
\begin{aligned}
\mathcal{L} &= \mathbb{E}_Q\left[g(X, Z; f)\right] + \mathbb{E}_P\left[\frac{dQ}{dP}(X, Z)(g(X, Z; f) - \ell(Y, X; f))\right] \\
&= \mathcal{L}_{\text{DR}}.
\end{aligned}
$$

### A.2. GLM Doubly Robust Objective

Recall the covariate shift assumption $\mathbb{E}_Q[Y \mid X, Z] = \mathbb{E}_Q[Y \mid X, Z]$ and that $g(X, Z; f) = \mathbb{E}[Y \mid X, Z]$.

$$
\mathcal{L} = \mathbb{E}_Q[-Y \cdot \eta(X; f) + b(\eta(X; f))].
$$

Since $\mathbb{E}_Q[b(\eta(X; f))]$ does not depend on the labels $Y$, we can focus on $\mathbb{E}_Q[-Y \cdot \eta(X; f)]$.

$$
\mathbb{E}_Q[-Y \cdot \eta(X; f)] = \mathbb{E}_P\left[-\frac{dQ}{dP}(X, Z)Y \cdot \eta(X; f)\right],
$$

and

$$
\begin{aligned}
\mathbb{E}_Q[-Y \cdot \eta(X; f)] &= \mathbb{E}_Q[-\mathbb{E}_Q[Y \mid X, Z] \cdot \eta(X; f)] \\
&= \mathbb{E}_Q[-g(X, Z) \cdot \eta(X; f)].
\end{aligned}
$$

So

$$
\mathcal{L}_{\text{DR}} = \mathbb{E}_Q[b(\eta(X; f)) - \eta(X; f)g(X, Z)] - \mathbb{E}_P[\alpha(X, Z)\eta(X; f)(Y - g(X, Z))].
$$

# B. GLM Log-Likelihood Losses

### B.1. Regression

Recall that we have $\eta(x; f) = f(x)$, where $f(x) \in \mathbb{R}$ is the scalar prediction from the model, and $b(\eta(x; f)) = \frac{1}{2}\eta(x; f)^2$. Then letting $Y \in \mathbb{R}$ denote the label, the doubly robust loss for regression is:

$$
\mathcal{L}_{\text{DR}} = \mathbb{E}_Q[f(X)^2 - 2f(X)g(X, Z)] - 2\mathbb{E}_P\left[f(X)\frac{\text{Pr}_Q(X, Z)}{\text{Pr}_P(X, Z)}(Y - g(X, Z))\right].
$$

## B.2. Binary Classification

Recall that we have $\eta(x; f) = f(x)$, where $f(x) \in \mathbb{R}$ is the logit for the positive class, and $b(\eta(x; f)) = \log(1 + \exp(\eta(x; f)))$. Then letting $Y \in \{0, 1\}$ denote the label, the doubly robust loss for binary classification is:

$$\mathcal{L}_{\text{DR}} = -\mathbb{E}_P \left[ \frac{\Pr_Q(X, Z)}{\Pr_P(X, Z)} \log \left( \frac{f(X)}{1 - f(X)} \right) (Y - g(X, Z)) \right] - \mathbb{E}_Q[g(X, Z) \log f(X) + (1 - g(X, Z)) \log(1 - f(X))].$$

## B.3. Multiclass Classification

Recall that we have $\eta(x; f) = f(x)$, where $f(x) \in \mathbb{R}^K$ is the logit vector for the $K$ classes, and $b(\eta(x; f)) = \log \sum_{j=1}^{K} \exp(\eta_j(x; f))$. Then letting $Y \in \{0, 1\}^K$ denote the one-hot label vector and $g_k(X, Z) = \mathbb{E}[Y_k \mid X, Z]$, the doubly robust loss for binary classification is:

$$\mathcal{L}_{\text{DR}} = -\mathbb{E}_P \left[ \sum_{k=1}^{V} \frac{\Pr_Q(X, Z)}{\Pr_P(X, Z)} \log f_k(X)(Y_k - g_k(X, Z)) \right] - \mathbb{E}_Q \left[ \sum_{k=1}^{V} \log f_k(X) g_k(X, Z) \right].$$

## B.4. Text Generation

Recall that we have $\eta(x; f) = f^{(t)}(x)$, where $f^{(t)}(x) \in \mathbb{R}^K$ is the logit vector for the $K$ tokens in the vocabulary, and $b(\eta(x; f)) = \sum_{t=1}^{T} \log \sum_{j=1}^{K} \exp(\eta_j^{(t)}(x; f))$. Then letting $Y^{(t)} \in \{0, 1\}^K$ denote the one-hot label vector for the next token at time $t$ and $g_k^{(t)}(X, Z) = \mathbb{E}[Y_k^{(t)} \mid X, Z]$, the doubly robust loss for binary classification is:

$$\mathcal{L}_{\text{DR}} = -\mathbb{E}_P \left[ \sum_{t=1}^{T} \sum_{k=1}^{K} \frac{\Pr_Q(X, Z)}{\Pr_P(X, Z)} \log f_k^{(t)}(X)(Y_k^{(t)} - g_k^{(t)}(X, Z)) \right] - \mathbb{E}_Q \left[ \sum_{t=1}^{T} \sum_{k=1}^{K} \log f_k^{(t)}(X) g_k^{(t)}(X, Z) \right].$$

# C. Experiments

## C.1. Language Model Implementation

*Table 2.* Technical details for language model implementations.

|  | Language | Library | Model |
|---|---|---|---|
| LIWC | Python | liwc | - |
| Empath | Python | empath | - |
| SenteCon | Python | sentecon | - |
| BERT | Python | transformers | bert-base-uncased |
| RoBERTa | Python | transformers | roberta-base |
| MiniLM | Python | sentence-transformers | all-MiniLM-L6-v2 |
| MPNet | Python | sentence-transformers | all-mpnet-base-v2 |
| GPT-4.1 | - | - | gpt-4.1 |

To implement our lexicons, we use the third-party `liwc` Python library and the `empath` library provided by its creators. SenteCon-LIWC and SenteCon-Empath representations are generated using the `sentecon` library. BERT and RoBERTa embeddings are obtained via the HuggingFace `transformers` library using the pre-trained models `bert-base-uncased` and `roberta-base`, respectively, while MPNet and MiniLM embeddings are obtained via the HuggingFace `sentence-transformers` library with the pre-trained models `all-mpnet-base-v2` and `all-MiniLM-L6-v2`. Finally, results from GPT-4.1 were retrieved through OpenAI's Batch API.

Additional technical details are provided in Table 2.

## C.2. Computing Resources

All experiments were conducted on consumer-level machines using consumer-level NVIDIA GPUs.

## D. Validating Benchmarked Sensitivity Parameters

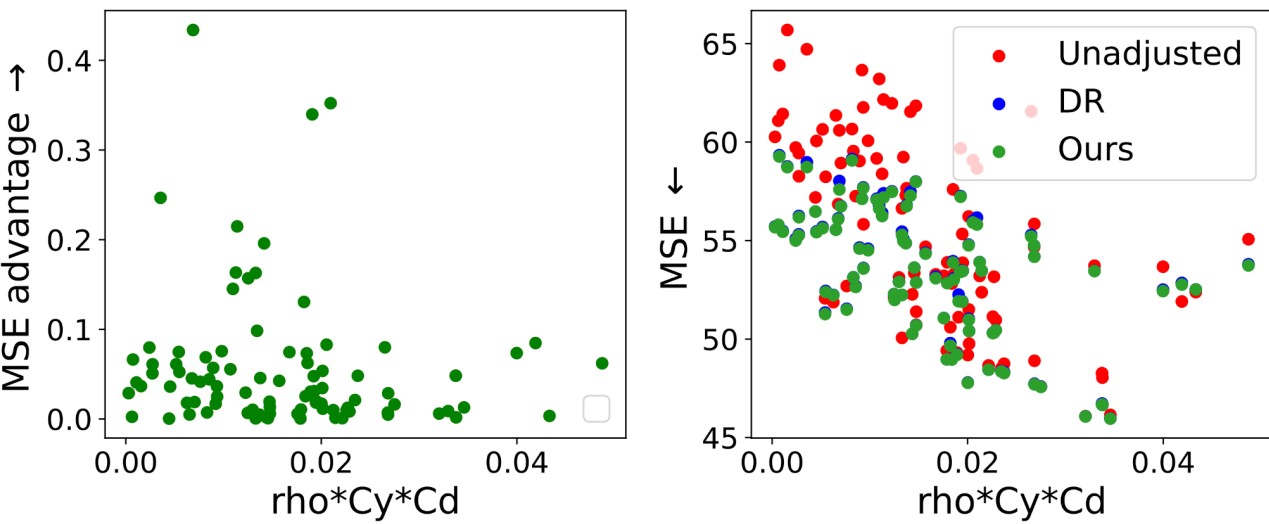

*Figure 6.* Amazon dataset with benchmarked sensitivity parameters. On the left, we plot the test performance difference (higher is better) between our approach and a model trained with a standard DR objective against the benchmarked sensitivity parameters. On the right, we plot the test performance of our approach and both baselines.

We conduct an additional experiment with the Amazon semi-synthetic dataset to provide evidence for the effectiveness of benchmarked sensitivity parameters. This experiment follows an identical setup to the Amazon experiment described in Section 5.3, with the sole difference being that instead of the true sensitivity parameters, we use benchmarked sensitivity parameters that we estimate by computing $C_Y, C_D, \rho$ between the short feature set and a *shorter* feature set where we further randomly drop half of the features from the short feature set. As shown in Figure 6, across 100 iterations, optimizing with our bound and these heuristically benchmarked sensitivity parameters offers equal or (typically) better performance compared to the standard DR loss.

# E. Additional Experimental Results

## E.1. Language Model Evaluation

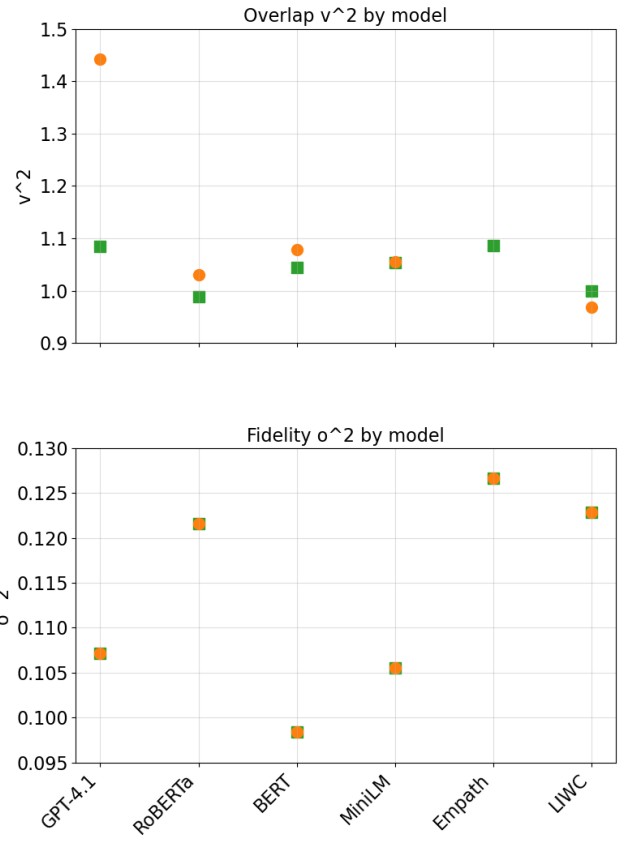

*Figure 7.* OVB metrics under distribution shift in the Math Reasoning dataset.

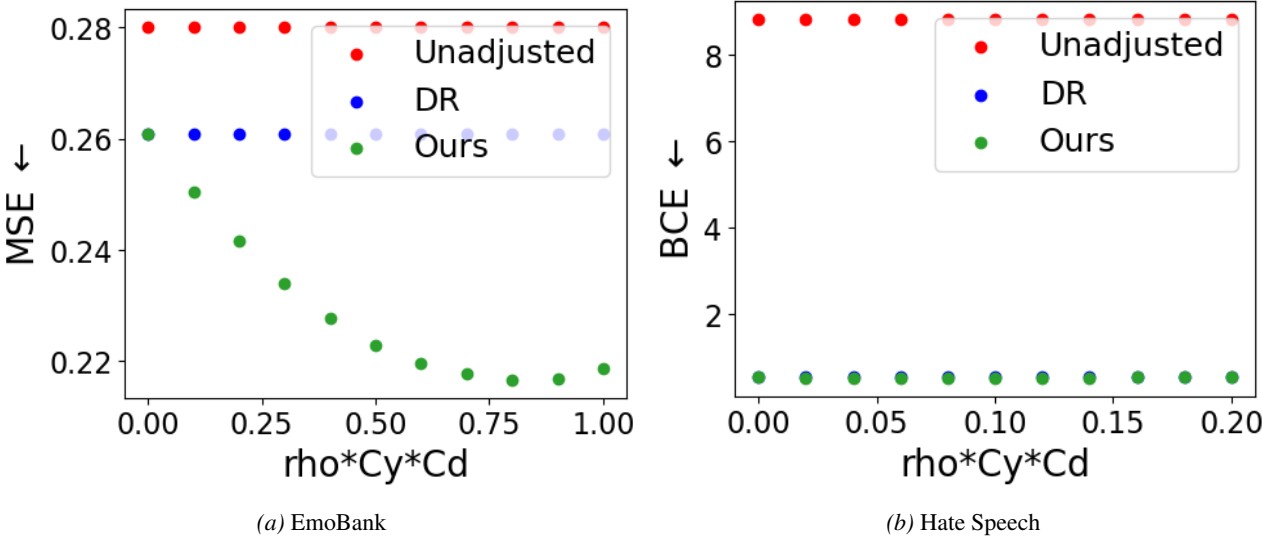

*(a)* EmoBank                                     *(b)* Hate Speech

*Figure 8.* Test performances of our approach, the DR baseline, and the unadjusted baseline plotted against strength of omitted variables.

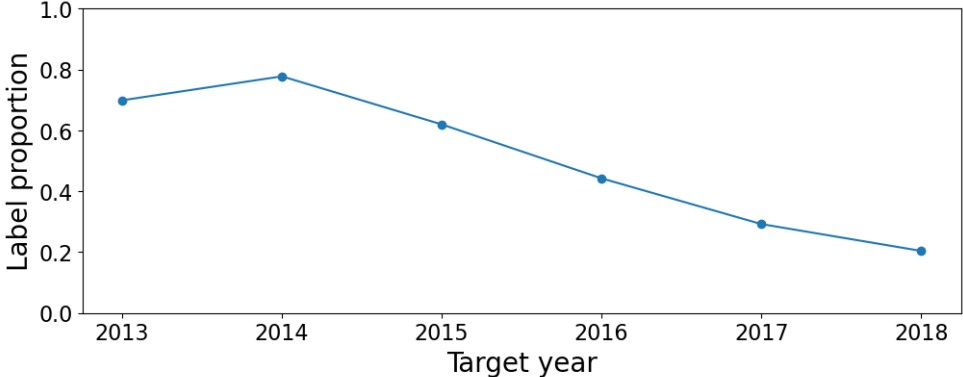

*Figure 9.* Wild-Time dataset. Proportion of labels in data splits corresponding to each year (Business $= 1$, Black Voices $= 0$).

