# OpenReview forum: "Omitted Variable Bias in Language Models Under Distribution Shift"
_ICML.cc/2026/Conference — ICML 2026 regular_

### Official Review · Reviewer_4So4 · 2026-02-28

**Soundness:** 4
**Presentation:** 4
**Significance:** 3
**Originality:** 3
**Overall Recommendation:** 5
**Confidence:** 2

**Summary:**

The paper argues that standard distribution-shift correction in language models addresses only observable shifts, neglecting unobservable factors arising from latent variables or uncaptured textual information. It formalizes this issue as omitted variable bias, derives a worst-case generalization bound via a doubly robust objective with sensitivity parameters, and evaluates the approach empirically. Experiments on comprehensive datasets show that standard DR is overly optimistic under distribution shift, while the proposed worst-case objective improves OOD performance with appropriate sensitivity calibration.

**Compliance With Llm Reviewing Policy:**

Affirmed.

**Final Justification:**

New results have addressed my concerns, and I will maintain my score.

**Key Questions For Authors:**

Can the framework be extended to end-to-end fine-tuning or preference optimization in modern LLMs, beyond feature-based linear or shallow MLP settings?

See Weakness

**Limitations:**

Yes

**Strengths And Weaknesses:**

### Strengths

**S1:**  The paper provides a clear and well-motivated problem formulation. The distinction between observable and unobservable distribution shift is compelling given that text must be compressed into finite-dimensional representations before modeling, making the central claim conceptually strong and relevant.

**S2:** It adapts omitted variable bias  to derive a worst-case generalization bound under distribution shift, with interpretable terms tied to fidelity, overlap, and sensitivity. The GLM specialization improves tractability under common NLP losses.

**S3:** Experiments span text generation, regression, and classification in both semi-synthetic and real-world settings, supporting the framework’s task generality.

**S4:** Using inferred sensitivity ranges to compare MATH and MATH-Perturb provides a novel perspective  into what aspects of the task distribution the model may be failing to capture.


### Weaknesses

**W1**: The method depends on sensitivity parameters that are not identifiable from observed data and must be calibrated, making its practical utility sensitive to user choices, especially in real-world settings.

**W2**: In the MATH study, sensitivity is tuned to match the true test performance. While analytically useful, this approach is less practical for real-world deployment when target labels are unavailable.

---

> ### Author Rebuttal · Authors · 2026-03-31
>
> Thank you for your many positive remarks noting the conceptual strength, empirical rigor, and broad task generality of our framework. We appreciate your additional comments and questions, which we address below.
>
> **[Sensitivity parameters not identifiable from data (W1)]** We agree that in optimization, practical deployment of our approach is sensitive to user choices regarding the sensitivity parameters. However, we emphasize that the unknowability of omitted variables is the precise motivation for a sensitivity analysis framework. Instead of computing a potentially biased point estimate for the target loss, parameterizing distribution shift over unobserved variables with $C_Y$, $C_D$, $\rho$ provides a rigorous basis for evaluation and model comparison (Sections 6.1 and 6.2).
>
> To anchor our sensitivity parameter choices, we reiterate that in optimization, plausible values for the sensitivity parameters can be benchmarked from the observed data (Section 5.3). In both EmoBank and Hate Speech, these benchmarked values fall within the “helpful” (i.e., performance-improving) range of sensitivity parameters.
>
> We provide new evidence of the effectiveness of benchmarked sensitivity parameters:
> - **New Amazon experiments** ([anonymous link](https://spoo.me/amazon-ovb) to results): Instead of the true sensitivity parameters for the Amazon experiment, we use benchmarked sensitivity parameters that we estimate by computing $C_Y, C_D, \rho$ between the short feature set and a *shorter* feature set where we further randomly drop half of the features from the short feature set. The experimental setting remains otherwise the same as in the paper. Across 100 iterations, optimizing with our bound and these heuristically benchmarked sensitivity parameters offers equal or (typically) better performance compared to the standard DR loss.
> - **Temporal drift experiments** ([anonymous link](https://spoo.me/huffpost-ovb) to results): In a new temporal distribution shift experiment, we observe that the benchmarked sensitivity parameters also fall within an optimal range for performance improvement relative to standard DR adjustment (more details of this experiment in our response to reviewer w1ZM titled “Experiment scope + temporal drift data”).
>
> While it is still possible for benchmarked sensitivity parameters to hurt model performance, our approach consistently matches or improves performance across our evaluation settings, even when the proxy models are constructed through a simple heuristic and benchmarked parameters do not perfectly align with the *maximum* possible performance improvement.
>
> **[Sensitivity tuned to match test performance for MATH (W2)]** We clarify that the purpose of the MATH experiment was to show the utility of our method as an analytical tool for measuring the strength of omitted variables in language models. Our method provides a useful diagnostic tool for comparing language models over a criterion other than pure performance, particularly black-box LLMs. For actual deployment when target labels are unavailable, we do not tune sensitivity parameters to the test set; instead, we use the method where we benchmark the sensitivity parameters to a proxy model. This allows users to define a range of worst-case bounds without needing target labels.
>
> **[End-to-end fine-tuning / preference optimization in modern LLMs (Q1)]** Yes, our framework can be extended to these settings.
> - The most common preference optimization algorithms, like DPO or PPO-based RLHF, use Bradley-Terry reward models, which are GLMs. Therefore, preference optimization is actually a particularly natural fit for our approach because it benefits from the more tractable GLM-specific formulation of our method.
> - In theory, it is possible to use our approach for end-to-end fine-tuning of LLMs. In practice, because updating the LLM weights causes the representations $h(X)$ will then change at every gradient step, the nuisance parameters $g,\alpha$ must then also be re-estimated at every step, which is computationally demanding.
> - In practice, the most straightforward and lightweight way to apply our framework to modern LLMs is to train a final robust adaptation layer over the frozen LLM embeddings (either for text generation or for the reward model). A reward model adapted in this way should be a better representation of actual preference, as it explicitly combats issues like unobserved distribution shifts between the training annotators and the target deployment population for the LLM. This robust reward model can then be used for inference-time alignment approaches, e.g., as a rejection sampling criterion.
>
> We hope our response thoroughly addresses your questions. Thank you again for your feedback and strong support!

---

> > ### Author Rebuttal · Reviewer_4So4 · 2026-04-01
> >
> > Thanks to these additional experiments. My concerns have been addressed, and I will keep my positive score.

---

> > > ### Author Response · Authors · 2026-04-04
> > >
> > > We are glad that our response fully addressed your concerns. Thank you for your feedback, engagement, and continued support of our work!

---

### Official Review · Reviewer_w1ZM · 2026-03-07

**Soundness:** 3
**Presentation:** 2
**Significance:** 2
**Originality:** 3
**Overall Recommendation:** 3
**Confidence:** 3

**Summary:**

This paper covers data distribution shift for LLMs, such as the difference between training data and test data. Standard distribution shift correction methods do not work when important variables that cause the shift are unobservable to the model. The authors utilize doubly robust estimation and show theoretically that omitted variable bias occurs which leads to optimistic estimates of target performance. The authors derive a worst-case generalization bound that quantifies how bad performance can degrade from unobserved variables. The authors also develop sensitivity parameters as interpretable metrics to highlight when a model cannot capture concepts between source and target distributions. In their experiments, they find that the worst case bound can recover target performance when the sensitivity parameters are chosen well.

**Compliance With Llm Reviewing Policy:**

Affirmed.

**Final Justification:**

These results have addressed some of my concerns. Further analysis of the experiments I recommend are still needed, so I will maintain my score.

**Key Questions For Authors:**

Can this be mitigated for LLaMA-3.1-70B?

Can Omitted Variable Bias be measured and drift distribution be mitigated under this framework for temporal settings (e.g. new named entities or dumps of data)?

**Strengths And Weaknesses:**

Strengths
Strong theoretical framework to derive the OVB bound utilizing causal inference. The GLM simplification in Section 4 help makes the optimization problem tractable for many common NLP tasks.
The semi-synthetic Amazon setup is well-designed for validation. By constructing labels from known features and deliberately withholding subsets at training time, the authors create a controlled environment where ground-truth sensitivity parameters are known allowing a cleaner test of whether the worst-case bound actually does what it claims than purely real-world data would permit.

Weaknesses
1. Sensitivity parameter selection is underspecified. The entire framework hinges on choosing the correct sensitivity parameters, but these cannot be identified directly from data. In practice, a user could easily choose values that are too optimistic or too pessimistic, and the EmoBank/Hate Speech results illustrate that getting this wrong actively hurts performance.
2. The authors assume that the label distributions in these datasets are conditioned on the same covariates. However, in many real NLP settings, label distributions shift in ways that covariate reweighting cannot fix. The paper does not discuss how severely violations of this assumption would affect the bounds.
3. Experiment scope is limited. Three of four datasets are either semi-synthetic (Amazon) or small-scale (EmoBank, Hate Speech). The MATH reasoning experiment only evaluates the bound rather than optimizing against it, so the claim that worst-case optimization improves generalization is only validated on comparatively simple tasks.
4. No study of temporal drift of datasets. This is often the most common type of data distribution drift in practice. The analysis of how data distribution drift can be mitigated or modeled is not found over time.

---

> ### Author Rebuttal · Authors · 2026-03-31
>
> Thank you for your positive remarks on the strength of our theoretical framework, utility of our GLM formulation, and rigor of our semi-synthetic experimental design! We address your questions and concerns below.
>
> **[Covariate shift assumption (W2)]** Our method indeed assumes the conditional label distribution is invariant given the complete set of covariates. While this is a strong assumption, it is standard and fundamental in much of the distribution shift literature and is the same assumption on which IPW-based methods rely. Our work does not attempt to solve *all* forms of shift, such as label shift. Instead, we ask a specific question within an established distribution shift setting: assuming covariate shift holds over the full unobservable set of variables, what happens given that models can only observe a partial lower-dimensional subset?
>
> **[Experiment scope + temporal drift (W3, W4, Q2)]** We thank the reviewer for this suggestion. Temporal drift is indeed a highly relevant form of distribution shift, particularly for modern language models. Therefore, following the reviewer's suggestion, we conduct an additional experiment specifically targeting temporal drift.
>
> We use the Huffpost dataset from the Wild-Time temporal distribution shift benchmark, where the task is to classify news headlines into categories and the distribution of categories changes over time. We look at drift from the year 2012 (source distribution) to 2018 (target distribution) and take headlines in the Black Voices and Business categories. We use MPNet embeddings as our language representation and benchmark our sensitivity parameters using a proxy “shorter” model built on simpler features from the lexicon Empath.
>
> Our results ([anonymous link](https://spoo.me/huffpost-ovb)) indicate that in this temporal drift setting, our approach consistently improves performance in the target distribution relative to standard DR adjustment. We will include this additional experiment to the final paper.
>
> **[Sensitivity parameters not identifiable from data (W1)]** As you point out, because sensitivity parameters cannot be identified from data, guessing values incorrectly can hurt performance. However, we emphasize that the unknowability of omitted variables is the precise motivation for a sensitivity analysis framework. Instead of computing a potentially biased point estimate for the target loss, parameterizing distribution shift over unobserved variables with $C_Y, C_D, \rho$ provides a rigorous basis for evaluation and model comparison (Sections 6.1 and 6.2).
>
> To mitigate the risk of choosing sensitivity parameters incorrectly during optimization, we reiterate that we benchmark plausible sensitivity parameters from the observed data (Section 5.3). In both EmoBank and Hate Speech, these benchmarked values fall within the performance-improving range of sensitivity parameters.
>
> We provide new evidence of the effectiveness of benchmarked sensitivity parameters:
> - **New Amazon experiments** ([anonymous link](https://spoo.me/amazon-ovb) to results): Instead of the true sensitivity parameters, we use benchmarked sensitivity parameters that we estimate by computing $C_Y, C_D, \rho$ between the short feature set and a *shorter* feature set where we further randomly drop half of the features from the short feature set. Across 100 iterations, optimizing with our bound and these heuristically benchmarked sensitivity parameters offers equal or (typically) better performance compared to the standard DR loss.
> - **Temporal drift experiments**: In the temporal drift experiments above, we observe that the benchmarked sensitivity parameters also fall within an optimal range for performance improvement relative to standard DR adjustment.
>
> While it is still possible for benchmarked sensitivity parameters to hurt model performance, our approach consistently matches or improves performance across our evaluation settings, even when the proxy models are constructed through a simple heuristic and benchmarked parameters do not perfectly align with the *maximum* possible performance improvement.
>
> **[Llama 3.1-70B (Q1)]** Yes, our approach can be used to account for OVB in modern LLMs like Llama 3.1-70B. The most straightforward and lightweight way to do so is to train a final robust adaptation layer $f$ over the frozen Llama embeddings $h(X)$ using the worst-case generalization loss $L_{OVB}$. In theory, full fine-tuning of an LLM with $L_{OVB}$ is also possible, but because updating the LLM weights causes the representations $h(X)$ to change at every gradient step, the nuisance parameters $g,\alpha$ must then also be re-estimated at every step, which is computationally demanding.
>
> We hope our response fully addresses your concerns and that you will consider raising your score. Thank you again!

---

> > ### Author Rebuttal · Reviewer_w1ZM · 2026-04-03
> >
> > These results have addressed most of my concerns. However, the details of the Amazon experiments and Temporal Drift experiments cannot be fully elaborated given the character limit of the response. Additionally, adding these results would take a significant amount of time, with the added further analysis of these experiments (does the drift correspond linearly with time? Does the smaller estimation model also degrade further over time, affecting performance?)
> >
> > I will maintain my score.

---

> > > ### Author Response · Authors · 2026-04-04
> > >
> > > Thank you for your response—we are glad our new results addressed most of your concerns.  We address your new comments and follow-up questions below.
> > >
> > > We understand your concern regarding the time required to perform a rigorous analysis of the temporal drift experiments. To ensure this question was answered during the discussion period, we prioritized this computation and **present an additional full year-by-year evaluation of the Wild-Time Huffpost dataset from 2013-2018**, where we show that our approach consistently mitigates temporal drift. We hope this comprehensive analysis fully resolves the concern.
> > >
> > > In this analysis, we keep 2012 (the first year in the dataset) as the source distribution and iterate through years 2013-2018 as the target distribution. For each iteration, we compute:
> > > - **Plot 1** ([anonymous link](https://spoo.me/uspzFI4)): Label proportion (Business = 1, Black Voices = 0), full model accuracy, and smaller proxy estimation model accuracy.
> > > - **Plot 2** ([anonymous link](https://spoo.me/NLkJhKy)): Performance of the unadjusted baseline, DR baseline, and our worst-case objective approach. For our worst-case objective, we set the sensitivity parameters to the benchmarked values, which we recompute for each year. As before, we benchmark these by comparing the full model (MPNet representations) with the proxy “shorter” model (lexical Empath features).
> > >
> > > ### **Does the drift correspond linearly with time?**
> > >
> > > As shown in plot 1, the overall label proportion generally changes linearly with time (a decreasing proportion of Business relative to Black Voices, which is consistent with overall societal trends), though not perfectly (e.g., an increase in 2014). This suggests a general trend of increasing temporal drift over time.
> > >
> > > This increasing drift over time is reflected in our optimization results in plot 2.
> > > The error of the unadjusted baseline shows a general upward trend, meaning that due to increasing distribution shift, a model trained on 2012 data is increasingly unable to perform on data that is farther in the future.
> > > Notably, the error of the standard DR baseline *also* shows a (slower) upward trend, indicating that there is also an increase in *unobservable* drift that the standard approach is unable to account for.
> > > **Our worst-case bound optimization continues to mitigate these severe shifts over time, with a growing advantage over the DR baseline.** As the drift grows over time, our approach’s performance advantage over the DR baseline generally grows, notably reducing the BCE loss by more than half during the large shifts in 2016 and 2018.
> > >
> > > We note that the performance advantage of our approach is not strictly linear, and in 2014 our method anomalously underperforms the DR baseline. We believe this to be an empirical example of the caveat regarding the benchmarking of sensitivity parameters that we mention in our prior response. As we previously stated, benchmarked sensitivity parameters do not always reflect *optimal* performance, and there can exist certain distributions where the proxy heuristic yields overly pessimistic parameters that are detrimental to performance.
> > >
> > > ### **Does the smaller estimation model also degrade further over time, affecting performance?**
> > >
> > > As seen in plot 1, the performance of the smaller proxy estimation model does indeed degrade over time at a faster rate than the full model. However, **this degradation is precisely what allows the benchmarking process to function effectively.** Because the proxy model is highly sensitive to distribution shift due to its less comprehensive feature set, then as its performance worsens over time relative to the full model, this scales the benchmarked sensitivity parameters to become more conservative (i.e., larger, which exactly reflects the increasing unobserved distribution shift over time). As the sensitivity parameters increase over time, this calibrates the bound to be more pessimistic, which allows the model to generalize successfully under increasingly severe distribution shifts.
> > >
> > > Therefore, rather than harming our method, the degradation of the proxy model over time is what allows us to maintain comparable performance on both the earliest target year (2013) and the latest target year (2018).
> > >
> > > ### **Addressing space constraints and the Amazon experiment**
> > >
> > > We hope this additional context also helps to address your concern regarding the space constraints of the rebuttal being insufficient to fully explain the temporal drift experiment. Regarding the new Amazon experiment, we clarify that the experimental setup is identical to the Amazon experiment described in the paper, other than the use of benchmarked sensitivity parameters instead of the true sensitivity parameters.
> > >
> > > We again thank you for your engagement with our work. We hope this response has fully addressed your concerns and that you will consider raising your score!

---

### Official Review · Reviewer_5xyv · 2026-03-13

**Soundness:** 2
**Presentation:** 2
**Significance:** 2
**Originality:** 3
**Overall Recommendation:** 3
**Confidence:** 4

**Summary:**

the paper adapts ideas from sensitivity analysis / omitted variable bias and derives a worst-case generalization bound for target-domain loss under unobserved confounding. The resulting bound augments a doubly robust estimate with a penalty term depending on estimable quantities (such as prediction error and overlap-like terms) and several user-specified sensitivity parameters capturing the strength of omitted-variable effects. The paper then uses this bound for three purposes: (i) more conservative evaluation under shift, (ii) robust training by optimizing the worst-case bound, and (iii) interpreting how strong omitted variables would need to be to explain observed degradation.

**Compliance With Llm Reviewing Policy:**

Affirmed.

**Key Questions For Authors:**

Concern 1: theoretical novelty relative to Chernozhukov et al.
A central question is how much of the theoretical development here is genuinely new relative to the omitted-variable-bias / sensitivity-analysis results in Chernozhukov et al. My current understanding is that the main derivation may largely adapt an existing OVB bound to the language-model distribution-shift setting, rather than developing a substantially new statistical argument. If that is the case, the paper’s main novelty may lie more in the problem formulation and reframing than in the core theory itself. I would encourage the authors to clarify more explicitly:
1. Which theorem or bound is essentially inherited from prior work?
2. What part is a nontrivial extension specific to the language-model / distribution-shift setting?
3. and whether the contribution is mainly conceptual, methodological, or theoretical?

Concern 2: role of doubly robust estimation from a semiparametric perspective
The paper relies on a doubly robust construction, but it is not yet clear to me what the semiparametric advantage is in this setting. In particular, I am unsure whether the proposed estimator is merely borrowing the DR form as a convenient decomposition device, or whether it actually enjoys the usual semiparametric benefits associated with DR estimation. For example:
1. Is the target parameter here pathwise differentiable under the proposed model?
2. Does the estimator correspond to an efficient influence-function-based construction?
3. Can it attain a semiparametric efficiency bound under correctly specified nuisance components?
4. If not, in what sense is the DR structure essential beyond robustness to nuisance misspecification?

**Limitations:**

see my question

**Strengths And Weaknesses:**

strength:
The work brings together ideas from econometrics / causal inference and ML robustness / domain shift in a clean way. The adaptation of OVB-style sensitivity analysis to language model evaluation is intellectually interesting and could open up useful dialogue between these communities.

weakness:
A central limitation is that the key sensitivity quantities controlling the OVB penalty are not identified from data. In practice, the user must specify them or calibrate them indirectly. This makes the method difficult to deploy in a principled way, and the paper does not fully resolve how a practitioner should choose these values.

---

> ### Author Rebuttal · Authors · 2026-03-31
>
> Thank you for your thoughtful feedback and for highlighting that our work bridges causal inference and ML robustness in a clean and interesting way! We address your questions and comments below.
>
> **[Sensitivity parameters not identifiable from data (W1)]** As you point out, sensitivity parameters by definition cannot be identified from the data. We emphasize that the unknowability of omitted variables is the precise motivation for a sensitivity analysis framework. Instead of computing a potentially biased point estimate for the target loss, parameterizing distribution shift over unobserved variables provides a rigorous basis for evaluation and model comparison (Sections 6.1, 6.2).
>
> To choose plausible sensitivity parameter values for optimization in practice, we reiterate the approach of benchmarking from the observed data (Section 5.3). In addition to the EmoBank and Hate Speech results, we provide new evidence of the effectiveness of benchmarked sensitivity parameters:
> - **New Amazon experiments** ([anonymous link](https://spoo.me/amazon-ovb) to results): Instead of the true sensitivity parameters, we use benchmarked sensitivity parameters estimated by computing $C_Y, C_D, \rho$ between the short feature set and a *shorter* feature set where we further randomly drop half of the features from the short feature set. Across 100 iterations, optimizing with our bound and these heuristically benchmarked sensitivity parameters offers equal or (typically) better performance compared to the standard DR loss.
> - **Temporal drift experiments** ([anonymous link](https://spoo.me/huffpost-ovb) to results): In a new temporal distribution shift experiment, the benchmarked sensitivity parameters also fall within an optimal range for performance improvement (more experiment details in our response to reviewer w1ZM titled “Experiment scope + temporal drift data”).
>
> While it is still possible for benchmarked sensitivity parameters to hurt model performance, our approach consistently matches or improves performance across our evaluation settings, even when the proxy models are constructed through a simple heuristic and benchmarked parameters do not perfectly align with the *maximum* possible performance improvement.
>
> **[Novelty vs. Chernozhukov et al. (Q1)]** Thank you for this question. The core OVB bound (e.g., page 3 “OVB bound definition”) is taken directly from the framework by Chernozhukov et al. Our nontrivial extensions translate this bound into a practical tool for evaluating and optimizing language models. We view our contributions as:
> - **1 (conceptual).** We identify that OVB is a fundamental source of brittleness for language models due to unobserved variables (a) external to the text and (b) omitted from language representations. We find that this is particularly tractable to study in the distribution shift setting.
> - **2 (methodological).** Leveraging a distributionally robust optimization framework, we turn the OVB bound—originally a tool for post-hoc sensitivity analysis of causal parameters—into a worst-case generalization *loss* that can be used directly to more robustly evaluate and optimize language models.
> - **3 (methodological).** As we mention in the paper, direct use of the Chernozhukov et al. bound in a generic language model loss requires the outcome model to predict the loss itself, which can be challenging to compute, particularly for modern complex language models. Our technical contribution reformulates the generic worst-case generalization loss into a more computationally tractable optimization problem for GLM-form losses, which encompass core NLP tasks like classification, regression, and text generation.
>
> We will update the paper to make these contributions and their relationship to Chernozhukov et al. more clear.
>
> **[DR estimation (Q2)]** Yes, our worst-case loss $L_{OVB}$ does retain the standard semiparametric advantages of doubly robust estimators. Following Chernozhukov et al., we use the terminology of Neyman orthogonal scores, which in our nonparametric setting correspond to the efficient influence functions of the target parameters (Chen et al., 2026).
> - The doubly robust loss estimator $L_{DR}$ is Neyman orthogonal w.r.t. perturbations in the nuisance parameters $g, \alpha$.
> - When the components of the OVB bound are estimated with debiased machine learning—as we do in our work—then $\sigma^2$ is Neyman orthogonal w.r.t. $g$, and $\nu^2$ can be also formulated such that it is Neyman orthogonal (Section 4 of Chernozhukov et al.).
> - Because every component of the worst-case generalization objective is Neyman orthogonal, it retains standard semiparametric efficiency and robustness guarantees.
>
> We hope our response fully addresses your concerns and that you will consider raising your score. Thank you again!
>
> ---
>
> Chen, Y., Kennedy, E. H., & Balakrishnan, S. (2026). On the Equivalence between Neyman Orthogonality and Pathwise Differentiability. arXiv preprint arXiv:2603.15817.

---

> > ### Author Rebuttal · Reviewer_5xyv · 2026-04-04
> >
> > Thank you for the clarification. I still concern about the gap between your work and Chernozhukov's, because now the theoretical bound and semiparametric result are corollaries of theirs.
> >
> > Can you give me more evidence your difference between yours and Chernozhukov's?

---

> > > ### Author Response · Authors · 2026-04-04
> > >
> > > Thank you for your response and for allowing us to further clarify the mathematical relationship between our framework and the bounds in Chernozhukov et al. We address your request for specific evidence of theoretical differences between our work and theirs below.
> > >
> > > ### **Difference 1: DRO formulation**
> > > Our first explicit divergence from Chernozhukov et al. is how the bound is mathematically formulated and used. Chernozhukov et al. establish the OVB bound as a post-hoc diagnostic tool for conducting sensitivity analysis on causal parameters—i.e., the target parameter is a causal parameter such as a causal effect. In Section 3, we provide a path for translating this bound to a language model loss, which is not explicitly a causal parameter. **Specifically, one of our novel contributions is formulating this as a distributionally robust optimization (DRO) problem, where we use the Chernozhukov et al. bound as a mathematical tool to construct the uncertainty set.** This allows us to parameterize a set of plausible target distributions using omitted variable strength, giving us an optimizable worst-case generalization loss for training language models.
> > >
> > > ### **Difference 2: GLM formulation**
> > > Our second difference is the mathematical decoupling required to actually optimize this DRO bound for language models (detailed in Section 4). If the Chernozhukov et al. bound is applied directly to a language model’s loss, the outcome model $g$ must predict the conditional expectation of the loss itself:
> > > $$g(X,Z;f) = \mathbb{E}[l(Y,X;f)|X,Z]$$
> > > Because the loss function $l$ depends on the language model $f$, this generic formulation requires the nuisance model $g$ to *simulate or predict the generative predictions of the primary model $f$*—which itself is being updated during optimization, meaning that a new $g$ must be learned for every update step. For modern LLM tasks like text generation, this is challenging, computationally demanding, and can introduce optimization instability during training.
> > >
> > > **Therefore, one of our primary technical contributions is the restructuring of this doubly robust objective for the family of GLM log-likelihood losses, which is what makes the optimization problem tractable.** As seen in Section 4.1, by leveraging the log-linear structure of GLMs, we shift the dependency on the primary model $f$ out of the outcome model $g$ and entirely into the Riesz representer $\alpha$ via the GLM natural parameter $\eta$, i.e.,
> > >
> > > $$\alpha(X, Z; f)=\frac{\text{Pr}_Q(X,Z)}{\text{Pr}_P(X,Z)}\eta(X; f)$$
> > >
> > > This specific reformulation yields:
> > >
> > > $$g(X,Z) = \mathbb{E}[Y|X,Z]$$
> > >
> > > Under our formulation, the outcome model $g$ simply predicts the label, without any dependency on the language model $f$. This means that this nuisance parameter (1) needs to be learned only once during optimization and (2) is fundamentally much easier to learn because the task is to predict a label rather than another model’s prediction. This non-trivial decoupling is what successfully bridges the theory in Chernozhukov et al. to tractable NLP optimization.
> > >
> > > We will make these contributions clear in the revised paper.
> > >
> > > We again thank you for your engagement with our work. We hope this response has fully addressed your concerns and that you will consider raising your score!

---

### Official Review · Reviewer_Mtah · 2026-03-15

**Soundness:** 3
**Presentation:** 3
**Significance:** 2
**Originality:** 3
**Overall Recommendation:** 4
**Confidence:** 4

**Summary:**

The authors adapt omitted variable bias bounds from causal inference to evaluate and optimize language models under distribution shift. They define a partially identified doubly robust objective where only a textual representation $h(X)$ is observable. The paper aims to provide robust target domain evaluations by bounding worst case generalization loss. The authors specialize their general objective to generalized linear model likelihoods and provide empirical results for text generation and classification.

**Compliance With Llm Reviewing Policy:**

Affirmed.

**Final Justification:**

The rebuttal and the new experiments addressed some of my concerns. The new Amazon results using benchmarked sensitivity parameters is a useful empirical evidence for the approximation method. However, my concern regarding the short Riesz representer overlap assumption remains partially unresolved.

**Key Questions For Authors:**

1. How do you justify the overlap assumption for the short Riesz representer given that text embeddings concentrate on narrow manifolds in $\mathbb{R}^d$ making the measures $P$ and $Q$ practically mutually singular?
2. How does the proposed estimator avoid overfitting when computing the empirical variance of the Riesz representer since the true Radon Nikodym derivative diverges almost everywhere?
3. Can you clarify the causal terminology when the task is predictive distribution shift rather than treatment effect estimation?How sensitive is the empirical performance on the datasets to the specific choice of the proxy long model?

**Limitations:**

Yes, the authors have adequately discussed limitations.

**Strengths And Weaknesses:**

Strengths

- The paper usefully translates the general doubly robust loss to specific task losses for generalized linear models.
- Estimating target domain performance without target labels is an important and recognized problem.
- The synthetic Amazon dataset experiments show that the method works when the true omitted variables are known.
- Applying econometric omitted variable bias bounds to language model evaluation is a creative approach to distribution shift.

Weaknesses

The paper relies on a covariate shift assumption combined with unobserved external variables $Z$ which is a strong and untestable assumption. The framing conflates omitted variables in a causal sense with unobserved confounders in a purely predictive domain adaptation sense.

One potential technical weakness is the definition of the short Riesz representer $\alpha(h(X))$. The framework requires that for any square integrable function $f$ the identity $\mathbb{E}_P[\alpha(h(X)) f(h(X))] = \mathbb{E}_Q[f(h(X))]$ holds. This structural requirement implicitly demands absolute continuity of the target measure $Q$ wrt the source measure $P$ over the representation space $\mathbb{R}^d$ and strictly requires a bounded density ratio. Due to the concentration of measure in high dimensions text embeddings do not distribute uniformly but concentrate on narrow manifolds. Under distribution shift the respective manifolds for $P$ and $Q$ rarely align causing the probability measures to become practically mutually singular. The Radon Nikodym derivative diverges to infinity or drops to zero almost everywhere.

The omitted variable bias bounds scale directly with the norm of the Riesz representer $\mathbb{E}_P[\alpha(h(X))^2]$. Because the distributions lack meaningful overlap in $\mathbb{R}^d$ this theoretical expectation explodes. When attempting to estimate $\alpha(h(X))$ empirically from finite samples the estimators inevitably break down. Optimization objectives designed to learn the Riesz representer will overfit to the sparse samples where artificial overlap appears to exist resulting in violently unstable or artificially small empirical variance estimates. The theoretical guarantee of the doubly robust bound no longer holds because the required overlap parameter is practically infinite in the context of modern language models. The practical utility of the method is also limited because the sensitivity parameters can't be identified directly from the data.

---

> ### Author Rebuttal · Authors · 2026-03-31
>
> Thank you for your positive remarks recognizing the importance of the problem, our creative approach to distribution shift, and the utility of our GLM formulation! We address your remaining comments below.
>
> **[Overlap violations (W3, Q1, Q2)]** Overlap is a core assumption for any density ratio-based distribution shift adjustment method (e.g., IPW losses). However, as you point out, overlap violations can be a concern for high-dimensional language embeddings. This in fact highlights a strong use case of our work:
> - **Q1**: Our framework estimates the (short) Riesz representer empirically, and $\nu^2=E_P[\alpha(h(X))^2]$ is then an empirical measure of overlap distinct from the theoretical quantity. Very large $\nu^2$ warns of a possible overlap violation, meaning $P$ and $Q$ are sufficiently different that it is not possible to robustly adjust for distribution shift (as we might erroneously attempt to do in a standard setting). The worst-case generalization bound will also be very wide, correctly indicating that little can be inferred about the model’s performance in $Q$.
> - **Q2**: Empirically, our results suggest that this is less of a problem than one would expect, in part because we constrain the hypothesis class of the Riesz representer through model choices and further smooth/regularize with sample splitting and Platt scaling for GPT-4.1.
>
> **[Covariate shift assumption (W1)]** While covariate shift is a strong assumption, it is likewise standard and common in the distribution shift literature and is the same assumption on which IPW-based methods rely. We clarify that our work does not *assume* the presence of unobserved external variables; instead, it *allows* for their presence by relaxing the strict covariate shift assumption (which assumes all distribution shift occurs over variables that are fully modeled from the text).
>
> **[Omitted variables vs. unobserved confounders (W2, Q3 pt 1)]** We do not view this as a conflation, but rather a useful parallel between the causal inference and domain adaptation literatures. In a causal sense, consider the text itself to be a treatment and the label to be an outcome. Distribution shifts can arise due to unobserved confounders that *cause* both certain texts to be more likely to be written and certain labels to be more likely. For instance, in the Hate Speech dataset, the user populations of Reddit and Gab differ in ways that likely cannot be fully captured in the text or text representations, but do fundamentally affect how users produce and engage with texts.
>
> **[Sensitivity parameters not identifiable from data (W4)]** It is true that sensitivity parameters by definition cannot be identified from the data. However, the unknowability of omitted variables is the precise motivation for a sensitivity analysis framework. Instead of computing a potentially biased point estimate for the target loss, parameterizing distribution shift over unobserved variables provides a rigorous basis for evaluation and model comparison (Sections 6.1 and 6.2).
>
> We reiterate that in optimization, plausible sensitivity parameter values can be benchmarked from the observed data (Section 5.3). In addition to the EmoBank and Hate Speech results, we provide new evidence of the effectiveness of benchmarked sensitivity parameters:
> - **New Amazon experiments** ([anonymous link](https://spoo.me/amazon-ovb) to results): Instead of the true sensitivity parameters, we use benchmarked sensitivity parameters estimated by computing $C_Y, C_D, \rho$ between the short feature set and a *shorter* feature set where we further randomly drop half of the features from the short feature set. Across 100 iterations, optimizing with our bound and these heuristically benchmarked sensitivity parameters offers equal or (typically) better performance compared to the standard DR loss.
> - **Temporal drift experiments** ([anonymous link](https://spoo.me/huffpost-ovb) to results): In a new temporal distribution shift experiment, the benchmarked sensitivity parameters also fall within an optimal range for performance improvement (more experiment details in our response to reviewer w1ZM titled “Experiment scope + temporal drift data”).
>
> While it is still possible for benchmarked sensitivity parameters to hurt model performance, our approach consistently matches or improves performance across our evaluation settings, even when the proxy models are constructed through a simple heuristic and benchmarked parameters do not perfectly align with the *maximum* possible performance improvement.
>
> **[Sensitivity to choice of proxy model (Q3 pt 2)]** The new Amazon experiments demonstrate that the benchmarking process is reasonably robust to the choice of proxy model. Because the proxy "shorter" models are defined by randomly subsetting features, they cover a wide range of possible proxies.
>
> We hope our response fully addresses your concerns. Thank you again, and we hope you will consider strengthening your support for our paper!

---

> > ### Author Rebuttal · Reviewer_Mtah · 2026-04-04
> >
> > I thank the authors for the rebuttal and the new experiments. The Amazon results using benchmarked sensitivity parameters provide useful empirical evidence for the approximation method.
> >
> > My theoretical concern regarding the short Riesz representer overlap assumption remains partially unresolved. The authors state that constraining the hypothesis class and applying regularization mitigates the issue empirically. Smoothing a fundamentally divergent density ratio doesn't restore the theoretical guarantees of the doubly robust bound. When the underlying measures are mutually singular the true overlap parameter is infinite.
> >
> > The causal framing also requires a bit more refinement. The authors suggest viewing text as a treatment and the label as an outcome. In the Hate Speech example the user platform acts as a domain indicator rather than an unobserved confounder in a standard causal graph. The text is typically a mediator or a proxy of the underlying user state, not  a treatment assigned to a unit. The authors should clean up the terminology to prevent conflation between causal effect estimation and domain adaptation.

---

> > > ### Author Response · Authors · 2026-04-04
> > >
> > > Thank you for your response and for recognizing that the new Amazon experiments provide useful empirical evidence for our approach. We address your remaining theoretical points below.
> > >
> > > ### **Theoretical overlap**
> > >
> > > We appreciate your rigorous feedback. We agree that in high-dimensional continuous spaces, if the underlying measures for $P$ and $Q$ are mutually singular, the true overlap parameter is infinite and empirical smoothing does not restore formal DR guarantees.
> > >
> > > We clarify that we do not claim that formal DR guarantees hold in the representation space $h(X)$. Our assumption of strict overlap is made with respect to the "true" text and confounder space $(X,Z)$ (Section 3.1).
> > >
> > > When we move from the theoretical $(X,Z)$ space to the practical $h(X)$ space, we view our estimator strictly as an empirical approximation. While—as you point out—overlap can be violated in this space if $h(X)$ is a high-dimensional representation, in our approach the capacity of the short Riesz representer is constrained through model choices, smoothing, regularization, etc. This effectively projects $h(X)$ into a restricted space where overlap is satisfied.
> > >
> > > **We do not claim that this projection restores DR guarantees.** Rather, our experiments demonstrate that this restriction provides a stable and useful empirical procedure. It successfully prevents the estimators from overfitting to sparse samples or violent variance instability, ensuring that the empirical overlap measure $\hat{\nu}^2$ remains a useful metric for generalization limits. If there *is* a lack of overlap with respect to this restricted space, it will empirically show up as a severely inflated $\hat{\nu}^2$, warning the practitioner that adjustment is impossible.
> > >
> > > We further acknowledge that if there is no overlap in the *true* space $(X,Z)$, the true sensitivity parameters $C_Y$ and $C_D$ would become infinite. We would not be able to catch this divergence empirically, as the true $(X,Z)$ are unobservable.
> > >
> > > We will revise our theoretical discussion to explicitly acknowledge this limitation. Moreover, we will clearly distinguish between the formal theoretical requirements and guarantees in $(X,Z)$ and the empirical nature of the bounds computed over the restricted hypothesis class of the $h(X)$ representations.
> > >
> > > ### **Causal framing**
> > >
> > > We clarify that our intention is not to frame our task as causal effect estimation. As we note in Section 2, our motivation is that “because most language tasks are not explicitly causal, the question of how to extend bounds... to language models becomes less clear." Our goal is to bridge this gap by **borrowing a concept that appears often in causal inference—omitted variable bias—to formalize the impact of unobserved variables in a predictive setting.** We recognize, however, that there may be ambiguity because of the strong association of OVB with the causal literature. **Following your suggestion, we will refine the language in our paper to more clearly make this distinction.**
> > >
> > > We further clarify that the use of the "text as treatment" analogy in the previous rebuttal was meant to connect our work with a history of literature in causal NLP that formalizes text as a treatment variable (e.g., Egami et al., 2018; Pryzant et al., 2020; Lin et al., 2025). This language does not appear in the paper itself.
> > >
> > > We again thank you for your constructive feedback and engagement with our work. We hope these clarifications address your remaining theoretical concerns, and that you will consider strengthening your support for our paper!
> > >
> > > ---
> > >
> > > Egami, N., Fong, C. J., Grimmer, J., Roberts, M. E., & Stewart, B. M. (2022). How to make causal inferences using texts. Science Advances, 8(42), eabg2652.
> > >
> > > Lin, V., Morency, L. P., & Ben-Michael, E. (2025, October). Isolated Causal Effects of Natural Language. In International Conference on Machine Learning (pp. 37919-37941). PMLR.
> > >
> > > Pryzant, R., Card, D., Jurafsky, D., Veitch, V., & Sridhar, D. (2021, June). Causal effects of linguistic properties. In Proceedings of the 2021 Conference of the North American Chapter of the Association for Computational Linguistics: Human Language Technologies (pp. 4095-4109).

---

### Decision · Program_Chairs · 2026-04-30

**Decision:**

Accept (regular)

**Comment:**

This paper studies the problem of evaluating and optimizing language models under distribution shift when important variables affecting the shift are unobserved. The authors adapt omitted variable bias (OVB) sensitivity analysis from econometrics and causal inference to derive a worst-case generalization bound for target-domain loss. The framework combines a doubly robust objective with sensitivity parameters that capture the possible strength of omitted variables, and the authors further develop a GLM specialization that makes the objective tractable for common NLP losses. Reviewers generally agreed that the paper addresses an important problem and presents an intellectually interesting bridge between econometric sensitivity analysis and distribution shift in language modeling. This is my opinion too.

Two reviewers expressed strong support for the paper, highlighting the conceptual clarity of the problem formulation, the methodological connection between OVB analysis and distribution-shift robustness, and the potential of the framework to open a useful line of research. These reviewers also found the empirical validation, including the semi-synthetic Amazon setup and the broader experimental coverage across tasks, to provide meaningful evidence for the approach. In particular, the controlled Amazon experiments were viewed as a convincing way to test whether the bound behaves as expected when the omitted variables are known.

The other reviewers raised several concerns, primarily related to theoretical assumptions and practical deployment. These include questions about the overlap assumptions underlying the Riesz representer, the extent to which the theoretical results extend beyond existing OVB bounds (e.g., Chernozhukov et al.), the need to select sensitivity parameters that cannot be identified from data, and the limited scope of some experiments. The authors’ rebuttal addressed a number of these issues constructively. In particular, the authors clarified the relationship to prior OVB theory, explained the role of the DRO formulation and GLM reformulation, and added new empirical evidence - including additional Amazon experiments and temporal drift analysis - to support the practical behavior of the framework. While some theoretical questions remain open, the discussion suggests that these concerns are partly intrinsic to sensitivity-analysis approaches rather than flaws specific to this work.

Overall, I view this paper as a thoughtful and technically grounded attempt to bring tools from sensitivity analysis and econometrics into the evaluation and training of language models under distribution shift. Although the framework relies on assumptions and user-specified parameters that limit immediate practical deployment, the authors clearly acknowledge these limitations and provide empirical evidence supporting the usefulness of the approach. Given the positive assessments from two reviewers, the constructive rebuttal that addressed several of the main concerns, and, importantly, the intellectual contribution of the paper, my overall assessment is supportive of acceptance.